# Energy choices to health outcomes: A multidimensional analysis of risk in BRICS via PMG-ARDL approach

**Funda Kaya[1], Liton Chandra Voumik[2], Mamunur Rashid[3], Salma Akter[2], Sayeem Hasan Khan[2], Mahdi Salehi[4], Konrad Kochański [5], Grzegorz Zimon [6]***

**1** Department of Environmental Health, Aydin Adnan Menderes University, Efeler, Aydin, Turkiye, **2** Department of Economics, Noakhali Science and Technology University, Noakhali, Bangladesh, **3** Department of Information Technology, School of Business & Technology, Emporia State University, Emporia, KS, United States of America, **4** Department of Economics and Administrative Sciences, Ferdowsi University of Mashhad, Mashhad, Iran, **5** Institute of Economics and Finance, University of Szczecin, Szczecin, Poland, **6** Faculty of Management, Rzeszow University of Technology, Rzeszow, Poland

\* gzimon@prz.edu.pl

**Data Availability Statement:** All data are in the paper.

**Funding:** The author(s) received no specific funding for this work.

## Abstract

This article employs a Panel Mean Group Autoregressive Distributed Lag (PMG-ARDL) approach to investigate the interaction between carbon dioxide ($CO_2$) emissions, Gross Domestic Product (GDP), fossil fuel, renewable energy consumption, trade, and their collective impact on life expectancy within the BRICS nations. The research reveals compelling findings. Notably, $CO_2$ emissions and trade openness exhibit negative and statistically significant impact on life expectancy. In contrast, GDP per capita and renewable energy consumption are positive and significant determinants of longer life expectancy. The nuanced outcomes underscore the complex interplay of economic, environmental, and social factors within the BRICS nations. The effects found by PMG-ARDL and FMOLS are very comparable, except for the trade openness' coefficients, which is the inverse. These findings hold significant implications for policy interpretation and sustainable development strategies. As nations struggle to balance economic growth and environmental improvement with public health, tailored interventions targeting $CO_2$ reduction, trade openness, renewable energy, and GDP growth can collectively contribute to longer life expectancy. In a broader context, this research contributes to the global discourse on sustainability, economic improvement, and health issue.

## 1. Introduction

In the modern era, many people appeal to enjoy a healthy and better quality of life in suitable environment. Life expectancy, popularly used as a proxy of people's health, is a crucial indicator in determining living standards, well-being, and human health across the globe. Thus, life expectancy has recently been seriously concerned for climate change issues, environmental degradation, and energy consumption that affect public health and living standards [1–3].

**Competing interests:** The authors have declared that no competing interests exist

Several studies prove that a nation's economic, demographic, and ecological conditions influence life expectancy [1, 4–6]. The BRICS countries, consisting of five emerging major economies: China, India, South Africa, Brazil, and Russia, make massive contributions to the world's energy usage and carbon emissions. According to the statistics compiled by World Bank, in 2021, China's average life expectancy will be 78.21 years, 72.75 years in Brazil, and 69.36 years in Russia. In contrast, the life span in South Africa and India will be 62.34 and 67.24 years, respectively. As aforementioned, life expectancy is subject to some changes over time for several reasons, such as the improvement of healthcare facilities, changes in lifestyle, advancement in living circumstances, energy utilization, and economic development.

Energy Consumption directly impacts economic and infrastructural development, influencing various aspects of health, well-being, and quality of life. However, greenhouse gases and other pollutants emissions often soar due to after-effects of higher energy usage, which further contribute to air and water pollution, enhance mortality rates, and shorten life expectancy. Energy usage boosted economic activity as well as elevated the levels of $CO_2$ emissions [7–10]. The procedure of generating energy, for instance burning, is detrimental to the natural environment since it releases $CO_2$, which harms the environment. Global warming and climate change have been investigated primarily due to carbon dioxide emissions and fossil fuel usage by human activity. The Global Energy Statistical Yearbook [11] reported that China emitted the highest carbon emissions worldwide from the BRICS nations, and its total carbon emissions were recorded at 28.8% of the global emissions in 2020, with India's emissions making up 7.3%, the third highest in the world. Besides, South Africa, Russia, and Brazil each experienced carbon emissions of 4.5%, 1.4%, and 1.3%, correspondingly. The BRICS countries' combined carbon emissions in 2020 accounted for 43.3% of all carbon emissions worldwide. Traditionally, $CO_2$ emissions contribute to air pollution, environmental hazards, and various deleterious effects on the health of individuals [12–14]. What is more, Fossil fuel consumption, especially in the guise of burning coal, natural gas, and oil, significantly negatively influences life expectancy [15–17]. The BRICS are the key energy users of oil, coal, and gas; consequently, fossil fuel energy consumption will remain within 2040 [18]. Regarding the ENERDATA report [19] the total amount of coal burned by the BRICS countries turned out to be 5,217 metric tonnes, the total oil usage was 1,138 metric tonnes, and the total gas consumption was 910 billion cubic meters. So, vast amounts of fossil fuels have improved temperatures and greenhouse gases that induce climate change [20]. Burning pollutants like fossil fuels is a vast source of contamination of air quality and greenhouse gas emissions, leading to various health problems and premature mortality [21]. However, renewable energy consumption is beneficial to mitigate the negative consequences of fossil fuel consumption and essential for better public health and life expectancy. Promoting the usage of renewable energy can aid in mitigating environmental deterioration and extending life expectancy [22]. The transition from non-renewable energy to renewable energy sources such as solar, wind, and hydroelectric power can reduce environmental pollution, leading to cleaner air, fewer diseases, and a longer life span [23].

Subsequently, life expectancy instigates a vital role in achieving stable economic growth [24–27]. In the past few decades, BRICS countries have achieved tremendous economic growth, but unequal income distribution and access to primary resources affect human health. According to the International Monetary Fund's (IMF) estimation, Brazil's GDP is projected to be $2.08 trillion in 2023, $3.76 trillion in India, and $19.37 trillion in China. Russia's GDP was estimated at $2.21 trillion in 2022, and South Africa's GDP is projected to be $399.01 billion in 2023. The BRICS are predicted to contribute more than 50% of global GDP within 2023, up from 31.5% in 2020 (International Monetary Fund, 2023) [28]. Several studies, for example, Gulis [29]; Kim [30]; Jafrin et al. [31]; Schwandt et al. [32], detected that the earnings per person were indisputably linked to the expectation of life. Children from wealthy families

tend to have better health and lower rates of infant and child mortality [33]. This research aims to elucidate the intricate interplay between GDP, renewable and nonrenewable energy consumption patterns, environmental impacts and their collective influence on health outcome within the BRICS countries. By incorporating both economic, energy, and non-economic social factors, this study seeks to provide a comprehensive assessment of the diverse elements shaping life expectancy at birth. This approach offers a more complete view of the contributors to extended lifespan. Beyond correlations, this research endeavors to probe into the causal links between life expectancy and the identified independent variables. This analysis enriches the comprehension of the underlying dynamics and potential pathways for policy intervention.

The contributions of this study to the literature are stated below. (i) First, life expectancy was used as the health outcome in this study. While estimating the factors affecting life expectancy, both economic factors and some non-economic social reasons are included in the econometric model. By employing life expectancy at birth as the focal health indicator, this research merges economic determinants with broader societal influences. This holistic perspective enhances the understanding of longevity determinants. (ii) Unlike previous studies, this research uniquely dissects the effects of renewable and nonrenewable energy consumption on life expectancy within a unified model. This novel approach offers fresh insights into the distinct contributions of these energy types. (iii) The application of both PMG-ARDL and FMOLS methodologies represents a significant contribution to this research. PMG-ARDL offers a robust framework for analyzing the long-term relationships between variables in panel data, allowing for a comprehensive understanding of the intricate dynamics shaping life expectancy in BRICS nations. Meanwhile, FMOLS provides an additional layer of validation, ensuring the reliability and consistency of our findings. This dual-method approach enhances the rigor and credibility of our study, providing valuable insights into the nuanced connections between economic, environmental, and health factors. (iv) In this study, the causality relationship between life expectancy and independent variables was also examined.

This study is designed into five distinct segments: subsequently, the introduction part, the related literature in section 2: "Literature review", the data and research method in section 3: "Methodology", the result analyses and explanation in section 4: "Results and Discussion", and the conclusion with a policy implication drawn in the section 5: "Conclusions" and 6: "Policy Recommendations". At last the limitations and future research in section 7: "Limitations and Future Research".

## 2. Literature review

Prior research on the assessment of carbon-dioxide emissions, income, and energy consumption on life expectancy has produced diverse results depending on the employed methodologies, timespans, and countries being considered.

### 2.1. $CO_2$ emissions and life expectancy

Numerous research has demonstrated the detrimental effects of carbon emissions on life expectancy. As better health is necessary for emerging countries, it is also crucial to comprehend the factors that affect one's natural life. Ahmad et al. [34] inspected the connection among socio-economic factors, health, and ambient quality by applying data in China from 1960–2014. According to the results, natural gas, coal, and oil were chosen as environmental quality determiners, and they determined that these contaminants had long-term adverse effects on health. Landrigan et al. [35] explained that the mortality rate was more significant in countries with low or middle incomes than higher-income countries due to larger $CO_2$

emissions. It was also suggested that higher pollution negatively affected life expectancy. Asongu [36] examined the impact of increasing carbon dioxide emissions on human health enhancement from 2000–2012 in 44 SSA countries. The investigation proved that $CO_2$ emissions were harmful to human health. Hill et al. [37] evaluated whether $CO_2$ emission had a ruinous effect on the longevity of life in 49 US states from 2000 to 2010. They found that $CO_2$ emissions imposed an unfavourable effect on people's health. Agbanike et al. [38] applied the ARDL bound test to explore the impact of environmental contaminations and life's duration from 1971 to 2014 in Nigeria. They identified that $CO_2$ emissions had a harmful effect on life expectancy.

Mohammed et al. [39] identified the linkage between economic growth, development, and health in the selected emitting nations. They discovered a strong connection between these factors. They suggested reducing $CO_2$ emissions to achieve better life expectancy. Bighlii et al. [40] inspected the association between healthcare expenditure, economic upswing, and environmental degradation in 36 Asian nations. The study discovered through applying panel data that investments in the health care of public and private sectors could lower $CO_2$ emissions, which created better environmental quality. Adebayo et al. [41] accessed health, ICT, and $CO_2$ connection for the top ten ICT countries from 1986 to 2019. They discovered a detrimental effect of $CO_2$ on the environment and life expectancy. Additionally, they showed that most nations indicated a beneficial effect of ICT in reducing $CO_2$ emissions. Rahman et al. [42] examined whether environmental deterioration posed a risk to human longevity over time. Their finding also revealed that the higher carbon emission would lead to a lower duration of life in major polluted nations. Mahalik et al. [43] ascertained the linkage between $CO_2$ emissions and life expectancy for 68 emerging nations from 1990–2017. They identified how $CO_2$ emissions deteriorated life expectancy. Hence, they observed a positive linkage between life expectancy and $CO_2$ emissions in 39 emerging countries. Adebayo et al. [44] empirically discovered the relationship between financial globalization and $CO_2$ emissions in G7 countries from 1970 to 2018. Most of the findings showed a negative effect of globalization on $CO_2$ emissions. They suggested that limiting $CO_2$ emissions for better life expectancy was crucial.

Polcyn et al. [45] used the CS-ARDL approach to evaluate the connection between health expenditure, energy consumption, $CO_2$ emissions, population size, and income on health in 46 Asian nations from 1997–2019. They found that higher investment in healthcare improved life expectancy. They also showed that $CO_2$ emissions were detrimental to health. Das and Debnath [46] employed ARDL bound technique to reinvestigate the connection between $CO_2$ emission and life expectancy from 1991–2018 in India. They discovered a strong impact of $CO_2$ emission on life expectancy. Consequently, $CO_2$ emissions bring down life expectancy.

## 2.2. GDP and life expectancy

Shah et al. [47] results indicate the association between GDP, Life expectancy, and economic growth in G7 Countries from 1960 to 2017. They demonstrated that a higher life expectancy was correlated with a larger GDP per capita income. They also included that a higher life expectancy led to higher spending on health, which affected per capita real income. Liu et al. [48] ascertained China's economic growth and life expectancy outcomes from 1960 to 2010. The findings revealed that higher economic advancement led to an extension in life expectancy. Asghar et al. [49] identified the linkage between economic expansion and life expectancy by employing the ARDL bounds method in Pakistan from 1972 to 2017. They disclosed a favourable impact of economic prosperity on life expectancy. Murthy et al. [50] discovered how per capita income affected life expectancy between 1992 and 2017 by observing D-8 nations. They outlined that income substantially affected human health as it improved life

expectancy. Hendrawati et al. [51] revealed the correlation between income and life span in ASEAN countries from 1988 to 2018. They explored the idea that income ultimately lengthened life expectancy. They also found that a higher living standard could raise life expectancy. Rahman et al. [52] used annual data from 1996 to 2019 to analyse the nexus of people's earnings with life expectancy in ANZUS-BENELUX countries. They found that income exerts a health-promoting effect. They also emphasized that economic growth improved health care. Azam et al. [53] explained how income influenced people's longevity from 1975–2020 in Pakistan. The findings of the empirical analysis ultimately concluded that income enhances life expectancy over the long term.

## 2.3. Energy consumption and life expectancy

Recently, related literature has included the effect of non-renewable energy on life expectancy due to extreme fossil fuel usage, which boosts the probability of mortality. Nadimi et al. [54] examined non-renewable energy, which negatively influences people's health in Japan. This impact depends on the cost of climate change in the future and various strategies for reducing environmental pollution. Hanif [55] investigated the linkage between energy sources and people's health using the generalized method of moments (GMM) technique in Sub-Saharan Africa. Using fossil fuels harms life expectancy in Sub-Saharan African nations as it increases the death rate. Martins et al. [56] reported the impact of fossil fuel consumption on health in European countries. They found that fossil fuels were the primary economic drivers in many regions, which also denoted a remarkable influence on human health. Koengkan et al. [57] inquired about how renewable energy usage could help diminishes outdoor air pollution and death rates. They discovered a considerable impact of fossil fuels depletion on people's mortality rate. Ibrahim [58] explored the association between income level, non-renewable fuel sources, life expectancy, and African carbon emissions. They identified fossil fuels as being the leading source of energy consumption. They also found that fossil fuels and carbon emissions decrease life expectancy. In OECD countries, Mujtabe and Sahazad [59] evaluated the association between air quality degradation, economic upturn, and human health. They revealed a causal relationship between renewable energy sources and healthcare spending eventually. Majeed et al. [60] reviewed a study in 155 economies to examine the fastening between renewable energy depletion and health outcomes using GMM in 155 nations. They empirically discovered that renewable energy sources were beneficial to a long-life span. Rodriguez [61] evaluated the nexus of air quality and life expectancy by assessing the need for renewable energy in European countries. They discovered that spending on renewable energy had a beneficial impact on life expectancy. Karimi et al. [62] employed quantile regression to examine the role of renewable energy resources on life expectancy for G-7 countries. They discovered that renewable energy consumption and spending on health improved life expectancy, while carbon dioxide emissions shortened it.

## 2.4. Trade and life expectancy

According to previous research, trade openness can influence life expectancy through many factors. Trade openness generates higher income, influencing economic growth and better life expectancy. Herzer [63] demonstrated the causal connection between trade and public health in 74 countries. They found trade openness's beneficial impact on life expectancy and infant death rates. Sakyi et al. [64] explored the linkage between the free trade zone and African social welfare. They discovered that trade openness influenced health as tax revenues from trade activities affected the government's expenditure on health services. Majeed and Qadir [65] evaluated the impact of trade openness on health in Pakistan. They concluded that trade

openness had an adverse influence on life expectancy. Novignon et al. [66] investigated the linkage between health and trade openness in sub-Saharan Africa. They explained that trade openness raised life expectancy and decreased infant mortality. Dithmer and Abdulai [67] applied a panel cointegration method to evaluate the connection between trade openness and children's health. They reviewed that trade lowered the children's average mortality rate. They also found that a nation with superior institutions and minimal corruption was essential for better health. Shafi & Fatima [68] reported the connection between $CO_2$ emissions, trade liberalization, and life expectancy in China. They revealed that trade hampered life expectancy and raised $CO_2$ emissions. They implied that renewable energy could offer safe environment and increase life expectancy. Bouchoucha [69] evaluated the linkage between trade openness, $CO_2$ emissions, and the lifespan in 49 African nations from 1990–2019. They disclosed a detrimental influence of trade openness on life expectancy. Additionally, the consequences of $CO_2$ on health outcomes are highly detrimental. Upon a comprehensive analysis of the existing literature, it is evident that only a limited number of research have examined the potential consequences of energy consumption, economic growth, and human activity on health outcomes, specifically within the BRICS countries. Furthermore, the relationship between economic growth and its impact on health outcomes still needs to be explored. Furthermore, there is a need to investigate the interconnection between $CO_2$ emissions, the utilization of renewable energy sources, the promotion of trade liberalization, and the impact on economic growth, all of which influence life expectancy. This study can provide insights into policy initiatives that prioritize human well-being, foster equitable economic development, safeguard the environment, and improve life expectancy. This study addresses the existing research gap by employing the PMG-ARDL approach. No research within the BRICS region has employed this methodology to assess the environmental implications. This section will discuss previous research conducted in the field, and the present work aims to address the gaps identified in these studies.

## 3. Methodology

### 3.1. Data

This study investigates the impact of GDP, trade, environmental pollution, renewable and non-renewable energy consumption on life expectancy in BRICS countries. In the study, annual data for the period 1990–2019 were used, and the World Bank database and Our World in Data were applied to obtain the data. The descriptive information of the variables is given in Table 1.

In the study, firstly, descriptive statistics of the series were discussed. The descriptive statistics of the variables are shown in Table 2. As seen in Table 2, the mean value of the *logLE* series is 1.827, 0.559 for the *logCO2* series, 3.588 for the *logGDP* series, 1.133 for the *logFOSS* series, 0.051 for the *logREC* series, and 1.575 for the *logTO* series. The variable with the highest maximum value is the *logGDP* series, while the lowest is the *logREC* series. Moreover, the series with the highest minimum value is *logGDP*, while the *logREC* series has the lowest minimum value. Also, the highest standard deviation belongs to the *logREC* series.

Graphs of the variables used in the study are presented in Fig 1. According to the graphs in Fig 1, China has the longest life expectancy among the BRICS countries. Russia is the country with the highest $CO_2$ emissions among the BRICS countries. As for income per capita, the distribution is more complex and closer to each other. Russia is the country with the highest fossil fuel consumption. Among BRICS countries Brazil consumes the most renewable energy. The trade openness chart is more complex, the data are closer to each other, and the highest fluctuation belongs to Russia.

**Table 1. Data description and sources.**

| Variables | Abbreviation | Details and Measurement | Source |
|---|---|---|---|
| Life expectancy | $logLE_{it}$ | Life expectancy, total (year) | WB (2023) [70] |
| $CO_2$ emissions | $logCO_{2it}$ | $CO_2$ emissions (per capita metric tons) | WB (2023) [70] |
| Real income | $logGDP_{it}$ | GDP per capita (constant 2015 US$) | WB (2023) [70] |
| Fossil fuels energy | $logFOSS_{it}$ | Fossil fuels consumption, per capita (kWh) | Our World in Data (2023) [71] |
| Renewable energy | $logREC_{it}$ | Renewable energy consumption, per capita (kWh) | Our World in Data (2023) [71] |
| Trade openness | $logTO_{it}$ | Trade (% of GDP) | WB (2023) [70] |

## 3.2. Theoretical framework

Smith & Dunt [72] postulated a link between various combinations of medical and non-medical inputs and their corresponding output, represented by Eq 1 in his health production function.

$$HO = f(M.E) \tag{1}$$

The initials "HO" stands for "health outcomes," "M" for "medical resources," and "E" for "everything" else, including non-medical, social, economic, and lifestyle aspects. It is hypothesized that as healthcare spending enhances (M), so will health outcomes. After a certain point, additional investment in health care is unlikely to yield a significant health improvement. Numerous epidemiological, demographics, and economic research [73, 74], among others, have proposed a wide range of non-medical, social, economic, and physical factors as potential predictors of health conditions.

Therefore, health production depends not just on the health system and its input of resources, but also on non-medical, social, economic, and physical factors. Arawomo et al. [75] employed Smith's & Dunt's [72] generalized version of the "health production function" to evaluate the dynamic relationship between economic growth, energy use, and infant mortality in SSA countries. This study has as its theoretical foundation Or's [76] health production function proposal. Or [76] categorized the non-medical elements into three main groups to facilitate discussion. These include the natural setting, individual choices, and societal dynamics.

This study used life expectancy as a dependent variable, $CO_2$ emissions, GDP, fossil fuel, renewable energy consumption, and trade openness as independent variables. The function form of the study is presented below in Eq 2:

$$LE = f(CO_2, GDP, FOSS, REC, TO) \tag{2}$$

**Table 2. Descriptive statistics.**

| | $logLE$ | $logCO_2$ | $logGDP$ | $logFOSS$ | $logREC$ | $logTO$ |
|---|---|---|---|---|---|---|
| **Mean** | 1.827 | 0.559 | 3.588 | 1.133 | 0.051 | 1.575 |
| **Median** | 1.832 | 0.623 | 3.754 | 1.152 | -0.023 | 1.624 |
| **Max.** | 1.891 | 1.164 | 4.006 | 1.795 | 0.891 | 2.043 |
| **Min.** | 1.732 | -0.188 | 2.723 | 0.378 | -0.995 | 1.180 |
| **Std. Dev.** | 0.037 | 0.398 | 0.375 | 0.403 | 0.577 | 0.174 |
| **Obs.** | 150 | 150 | 150 | 150 | 150 | 150 |

Source: Author's calculation.

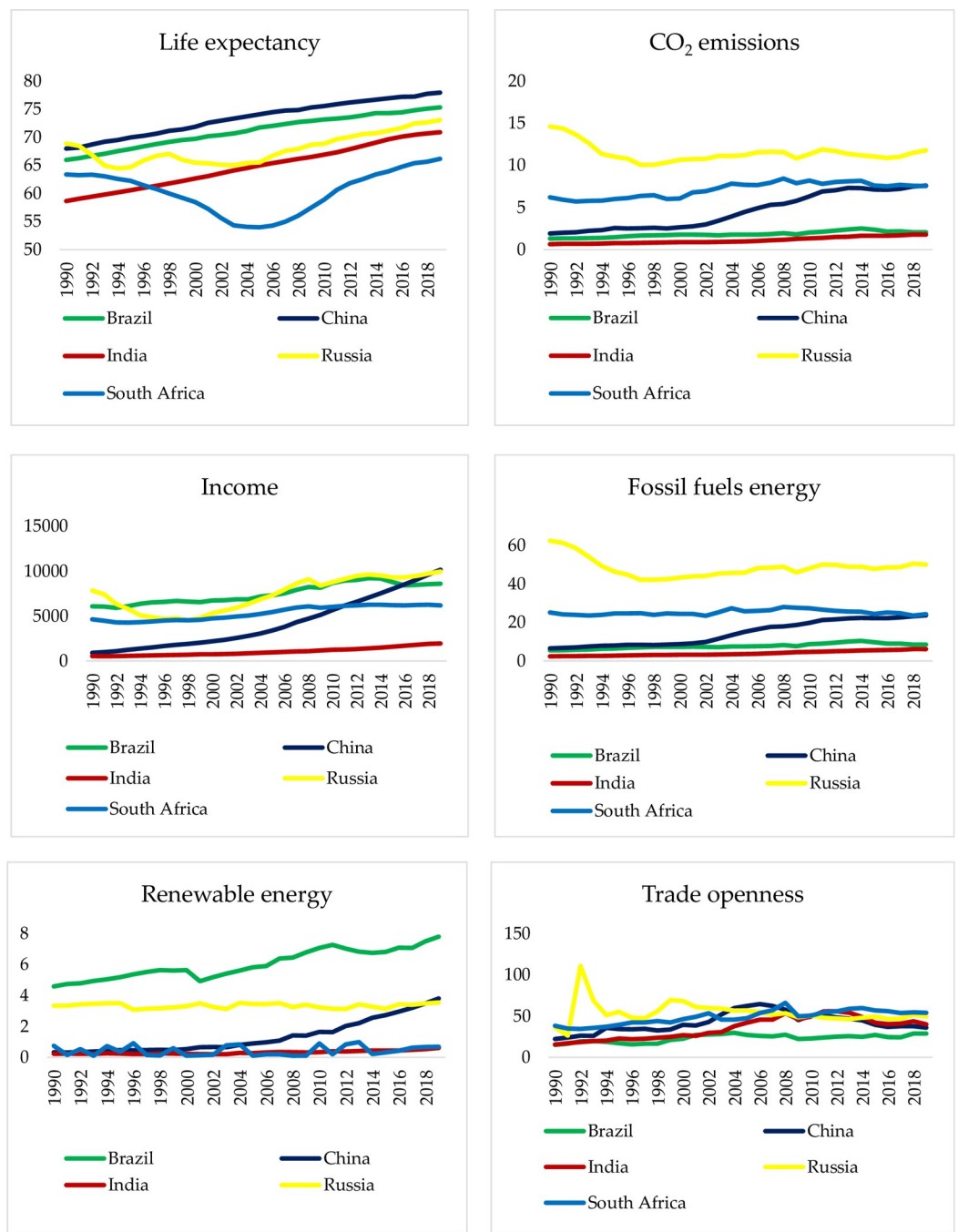

**Fig 1. Change of the series used in the study between 1990 and 2019.** Sources: World Bank (2023) [70] and Our World in Data (2023) [71].

In Eq (1), LE denotes the life expectancy (year), $CO_2$ emissions represent the carbon dioxide emissions, FOSS denotes the fossil fuel consumption, GDP represents the gross domestic product, REC stands for the renewable energy consumption, TO represents the trade openness.

Therefore, the model implemented during this investigation is defined in terms of the health-generating mechanism. This model has been specified in Eq 3 following the model

specification provided by Or [76].

$$LE_{it} = \alpha_i + M_{it}\beta + E_{it}\gamma + \mu_{it} \tag{3}$$

Torras [72] specifies energy consumption and income as the only relevant non-medical elements to consider in this study. Hence, M is a vector of medical variables assessed by trade openness, and E is a vector of non-medical factors typically referred to as ecological variables such as $CO_2$. Coefficients on M are denoted by $\beta$ and those on E by $\gamma$. It is important to note that $\gamma$ can be positive or negative, and $\beta$ that can be greater than zero. Health-related subjects of interest are linked in the study. Eq 4 provides a more precise form of health, as mentioned above equation:

$$
\begin{aligned}
logLE_{it} = \alpha_0 &+ \alpha_1 logCO_{2it} + \alpha_2 logGDP_{it} + \\
&\alpha_3 logFOSS_{it} + \alpha_4 logREC_{it} + \alpha_5 logTO_{it} + \mu_{it}
\end{aligned}
\tag{4}
$$

Here, symbolizes $\alpha_0$ is the constant term, $\alpha_{1-5}$ are coefficients, i represents the cross-section size, t is the time dimension, and $\mu_{it}$ is the error term.

The variables used in the study were transformed into logarithmic form. When applied to a distribution, the logarithmic adjustment can assist in bringing it closer to the symmetry and normality of a normal distribution. To improve the accuracy of statistical tests, skewed variables might be transformed into normally distributed ones. The logarithmic transformation may additionally be employed to control the variance of a variable, making the analysis more robust against outlying values.

## 3.3. Empirical methods

After the data were organized and defined, the cross-section dependency test was applied to the series' first in the study. This study uses the Pesaran CSD test for cross-section [77]. Misleading findings, size distortion, consistency bias, and cointegration are just some of the problems that have arisen because of cross-sectional dependence, which needs to be fixed [78–80]. Cross-section dependency testing is crucial when choosing the first-generation or the second-generation unit root tests. The cross-section dependency test should be applied for the results to be dependable and consistent. The following Eq 5 is used for the Pesaran CSD test:

$$CD_{test} = \sqrt{\frac{2T}{N(N-1)}} \left( \sum\nolimits_{i=1}^{N-1} \sum\nolimits_{k=i+1}^{N} \widehat{\tau_{ik}} \right) \tag{5}$$

The second-generation unit root tests should be preferred when cross-sectional dependence is detected between series. This study uses the CIPS unit root test for stationarity testing. The CIPS [81] unit root test is explained by the following Eq 6:

$$CIPS(N, T) = N^{-1} \sum\nolimits_{i=1}^{N} t_i(N, T) \tag{6}$$

This study used the PMG-ARDL method to investigate the long-term effects of $CO_2$ emissions, real income, fossil fuel and renewable energy consumption, and trade openness on life expectancy. The PMG-ARDL method was proposed by Pesaran et al. [82]. The PMG-ARDL method allows the variables to be stationary at various levels. It is used when the series becomes stationary at I(0), I(1), or a combination of both. The PMG considered a reduced level of heterogeneity, as this enforces homogeneity in the long-term projections and diversity in the short-term estimates. Moreover, it is noteworthy that the PMG can be applied in situations where certain variables exhibit stationarity at the level while others exhibit stationarity at

the first difference. Eq 7 defines the estimation of the PMG-ARDL panel paradigm outlined below:

$$\Delta Y_{1it} = \alpha_{1i} + \beta_{1i} Y_{1it-1i} + \sum_{l=2}^{k} \beta_{1i} X_{1it-1} + \sum_{j=1}^{p-1} Y_{1ij} \Delta Y_{1it-j} +$$
$$\sum_{j=0}^{p-1} \sum_{l=2}^{k} Y_{1ij} \Delta X_{1it-j} + \mu_{it} \tag{7}$$

Here, symbolizes $Y_1$; dependent variable, $\Delta X_1$; independent variable, $\mu_{it}$; error term, and $\Delta$; the first difference operator.

The paper applied the Dumitrescu Hurlin (D-H) panel causality test for the causality relationship between the series.

To test for causality, D-H causality calculate the individual Wald statistics ($W_{i,t}$) for the cross-section, and then take their arithmetic average and calculate the Wald statistics $\left( W_{N,T}^{HNC} \right)$. D-H causality [83] recommend using test statistics with an asymptotic distribution when $T > N$ and using test statistics with a semi-asymptotic $\left( Z_N^{HNC} \right)$ distribution when $T < N$. The following Eqs 8 and 9 are employed for the D-H causality test:

$$Z_{N,T}^{HNC} = \left( \sqrt{\frac{N}{2K}} W_{N,T}^{HNC} - K \right) \tag{8}$$

$$Z_N^{HNC} = \frac{\sqrt{N} \left[ W_{N,T}^{HNC} - N^{-1} \sum_{i=1}^{N} E\left( W_{i,T} \right) \right]}{\sqrt{N^{-1} \sum_{i=1}^{N} var\left( W_{i,T} \right)}} \tag{9}$$

## 4. Results and discussion

In the study, a slope heterogeneity test was applied to the series. Table 3 shows the results of the analysis. The p-values show strong evidence against the null hypothesis. In summary, significant slope heterogeneity was detected because of both tests. It is possible to say that the relationship between the variables discussed in the study varies between different groups and subpopulations.

This study then applied the Pesaran CSD [81] cross-sectional dependence test to test the cross-sectional dependence between the series. Table 4 shows the Pesaran CSD test results. The p-value of all variables in the Pesaran CSD test results is 0.000. This shows strong evidence against the null hypothesis. The results show that there is a cross-section dependence for all series. It shows that the series in this data set are not independent but that common factors are mutually influenced and influenced by each other.

After determining the cross-section dependency in all series, we applied the CIPS unit root test, which is one of the second-generation unit root tests to test the stationarity of the series. CIPS unit root test results are presented in Table 5. It is seen that $logLE$, $logCO_2$, $logGDP$, and

**Table 3. Slope heterogeneity test results.**

| Test | Value | p-value |
|---|---|---|
| $\Delta$ | 11.840*** | 0.000 |
| $\Delta_{adj}$ | 13.522*** | 0.000 |

Source: Calculated by the author. *** denotes significance at the 1% level.

**Table 4. CSD test results.**

| Variables | Pesaran CSD |
|---|---|
| $logLE$ | 10.328(0.000) *** |
| $logCO_2$ | 7.319(0.000) *** |
| $logGDP$ | 15.699(0.000) *** |
| $logFOSS$ | 5.832(0.000) *** |
| $logREC$ | 6.305(0.000) *** |
| $logTO$ | 8.029(0.000) *** |

Source: Calculated by the author. *** denotes significance at the 1% level.

*logREC* series become stationary after taking the first difference. In addition, it has been determined that the *logFOSS* and *logTO* series are stationary at I (0), that is,level. CIPS unit root test results became stationary after taking all variables' levels and the first difference in constant and constant + trend. In summary, it is supported by the results that the series has a constant mean and a stable behavior over time.

The PMG-ARDL estimator is the baseline regression to determine the long-term impact of $CO_2$ emissions, real income, fossil fuel, renewable energy consumption, and trade openness on life expectancy. The series is stationary at different levels is one reason for choosing the PMG-ARDL method. Table 6 shows the long-term results of PMG-ARDL. In the PMG-ARDL test results, it has been determined that $CO_2$ emissions have a negative and statistically significant effect on life expectancy in the long term. Results showed that a 1% increase in $CO_2$ emissions will reduce life expectancy by 0.451%. Also, having a relevant effect at the 1% significance level implies solid empirical evidence. Raihan et al. [84]; Nwani et al. [85]; Nkalu and Edeme [86]; Majeed and Ozturk [87]; Nica et al. [88]; Sultana et al. [89]; Alpopi et al. [90]; Nica et al. [91]; Das and Debanth [56]; and Polcyn et al. [45] demonstrate further that life expectancy is reduced in various regions due to CO2 emissions. BRICS countries are among the emerging economies and releasing massive amount of $CO_2$. The increase in $CO_2$ emissions can cause environmental pollution and deterioration of individual and public health. In addition, the increase in $CO_2$ emissions is among the most important causes of air pollution.

It has been determined that per capita income positively affects life expectancy in the long run and at the 1% significance level. It has been determined that a 1% increase in income increases life expectancy by 0.639%. The increase in the income per capita of individuals in the BRICS countries will affect their lifestyles, spending priorities, and investments in health. At this point, an increase in the income per capita may prolong people's life expectancy. Murthy

**Table 5. CIPS unit root test results.**

| Variable | Constant | | Constant +Trend | |
|---|---|---|---|---|
| | Level | 1st difference | Level | 1st difference |
| | CIPS unit root | | | |
| $logLE$ | -0.877 | -3.072*** | -2.936** | -3.284*** |
| $logCO_2$ | -2.258 | -3.435*** | -1.393 | -3.621*** |
| $logGDP$ | -2.179 | -2.844*** | -1.734 | -3.347*** |
| $logFOSS$ | -2.404** | -3.632*** | -1.754 | -3.855*** |
| $logREC$ | -1.931 | -5.582*** | -2.790 | -5.769*** |
| $logTO$ | -2.699*** | -5.078*** | -2.651 | -5.234*** |

Source: Author's calculation. *** and ** refer to significance at 1% and 5% levels.

**Table 6. PMG-ARDL long-run result.**

| Variables | Coefficient | t-Statistic | Prob. |
|---|---|---|---|
| $logCO_{2it}$ | -0.451*** | -4.902 | 0.000 |
| $logGDP_{it}$ | 0.639*** | 21.529 | 0.000 |
| $logFOSS_{it}$ | 0.001 | 0.011 | 0.990 |
| $logREC_{it}$ | 0.040*** | 4.397 | 0.000 |
| $logTO_{it}$ | -0.113** | -2.369 | 0.019 |

Source: Author's calculation. *** and ** show significance at the 1% and 5% levels.

et al. [50] investigated the relationship between income and life expectancy in D-8 countries. Through empirical analysis, they found that income increases life expectancy, like our results. Popescu et al [92], Kliestik et al. [93], Ebenstein et al. [94], Messias [95] and Rahman et al. [2018] [96] also demonstrated that rising income accelerated longevity.

The effect of fossil fuel consumption on life expectancy is not statistically significant. In addition, the coefficient is relatively low. It has been determined that fossil fuel consumption does not significantly affect life expectancy in the BRICS countries. Renewable energy has been shown to boost life expectancy in numerous studies, including those by Popescu et al [97], Nica et al [98], Lelieveld et al. [99] and Zimon et al. [100]. The effect of renewable energy consumption on life expectancy is positive and statistically significant. In other words, a positive long-term relationship exists between life expectancy and renewable energy consumption. A 1% increase in renewable energy consumption increases life expectancy by 0.040%. In addition, the significance level of 1% implies strong evidence for the statistical significance of this relationship. When obtaining renewable energy from clean and self-renewing sources and consuming these resources in BRICS countries will contribute to positive results on health. Therefore, this situation shows a curative effect on life expectancy. Renewable energy has been shown to boost life expectancy in numerous studies, including those by Rahman & Alam [101], Popescu et al. [102], Popescu et al. [103], Ghosh et al. [104], Rahman & Alam [42], and Liu & Zhong [26]. The effect of trade openness on long-term life expectancy is negative and statistically significant. It has been determined that there is a negative and long-term relationship between trade openness and life expectancy. A 1% increase in trade openness reduces life expectancy by 0.113%. Fostering trade openness was found to shorten life expectancy, as stated by Bouchoucha [69], Nica [105], Anderi et al. [106], Alam et al. [107], and Govdeli [108]. This finding contradicts Shafi & Fatima [68] which demonstrated that trade expansion accelerated life expectancy.

This study, FMOLS long-term coefficient estimator was used to test the robustness of the PMG-ARDL test results. FMOLS test results are listed in Table 7. We discovered that the $logCO_2$ series had a negative and statistically significant effect on the $logLE$ series. We found that the effects of the $logGDP$, $logFOSS$ and $logREC$ series on life expectancy were like the PMG-ARDL test results. However, we discovered that the $logTO$ series had a negative and significant effect at the 1% significance level on the $logLE$ series.

After the PMG-ARDL and FMOLS test results, this study used the D-H panel causality test to examine the causality relationship between. Table 8 gives D-H panel causality test results.

In the D-H panel causality test results, $H_0$ was rejected, and a bidirectional causality relationship was detected between $CO_2$ emissions and life expectancy. These results show that $CO_2$ emissions and life expectancy are caused by each other. Similarly, another bidirectional causal relationship identified between income and life expectancy. The income per capita and life expectancy in BRICS countries is caused by each other. A bidirectional causality has been

**Table 7. FMOLS test results.**

| Variables | Coefficient | t-Statistic | Prob. |
|---|---|---|---|
| $logCO_{2it}$ | -0.262*** | -6.107 | 0.000 |
| $logGDP_{it}$ | 0.102*** | 7.567 | 0.000 |
| $logFOSS_{it}$ | 0.194*** | 4.499 | 0.000 |
| $logREC_{it}$ | 0.023*** | 4.817 | 0.000 |
| $logTO_{it}$ | 0.048*** | 5.012 | 0.000 |

Note: *** show significance at the 1% levels.

found between fossil fuel consumption and life expectancy. However, we discovered that there was a unidirectional causal relationship from life expectancy to renewable energy consumption. We determined that there was a unidirectional causality relationship from trade openness to life expectancy.

## 5. Conclusions

Finding the impact of different drivers on life expectancy in the BRICS nations, this study set out to investigate the relationship between $CO_2$ emissions, GDP, fossil fuel, renewable energy, and trade openness. Employing the PMG-ARDL and FMOLS approaches, this research unveiled compelling insights into sustainable development and policy formulation within BRICS. $CO_2$ emission alleviated life expectancy by 0.451% for 1% enhanced. Trade openness also weakened life expectancy by 0.113%. The adverse impact of $CO_2$ and trade openness on life expectancy emphasizes the critical need for policies that balance economic development with environmental conservation. The positive influence of GDP and renewable energy signifies the potential of economic advancement and cleaner energy sources to enhance life expectancy. The GDP and renewable energy enhanced life expectancy by 0.639% and 0.040%, correspondingly, with a 1% boost in GDP and renewable energy. Unexpectedly, the increasing role of fossil fuel consumption shows positive impact on life expectancy, the findings is significant for FMOLS method and insignificant for PMG-ARDL.

D-H panel causality test was used to explore the causality relationship between the dependent and independent variables. Empirical evidence showed a bidirectional causal relationship between $CO_2$ emissions, the income per capita and fossil fuel consumption, and life

**Table 8. D-H panel causality test results.**

| No | Null hypothesis ($H_0$) | W- Stat | Zbar- Stat | Prob. | Causality |
|---|---|---|---|---|---|
| 1 | $logCO_2 \neq logLE$ | 9.584 | 11.639 | 0.000*** | $logCO_2 \rightarrow logLE$ |
| 2 | $logLE \neq logCO_2$ | 2.659 | 2.158 | 0.030** | $logLE \rightarrow logCO_2$ |
| 3 | $logGDP \neq logLE$ | 9.190 | 11.099 | 0.000*** | $logGDP \rightarrow logLE$ |
| 4 | $logLE \neq logGDP$ | 8.286 | 9.862 | 0.000*** | $logLE \rightarrow logGDP$ |
| 5 | $logFOSS \neq logLE$ | 13.391 | 16.851 | 0.000*** | $logFOSS \rightarrow logLE$ |
| 6 | $logLE \neq logFOSS$ | 4.761 | 5.036 | 0.000*** | $logLE \rightarrow logFOSS$ |
| 7 | $logREC \neq logLE$ | 0.769 | -0.429 | 0.667 | None |
| 8 | $logLE \neq logREC$ | 3.457 | 3.251 | 0.001*** | $logLE \rightarrow logREC$ |
| 9 | $logTO \neq logLE$ | 3.631 | 3.488 | 0.000*** | $logTO \rightarrow logLE$ |
| 10 | $logLE \neq logTO$ | 1.051 | -0.043 | 0.965 | None |

Source: Author's calculation. *** and ** show significance at the 1% and 5% levels.

expectancy. In addition, from life expectancy to renewable energy consumption a unidirectional causal relationship has been identified from trade openness to life expectancy.

The results strengthen economic, environmental, and social interdependence in shaping the environment, development, and health interconnection. As governments and policymakers navigate the intricate challenges of sustainable growth, the results of this study offer valuable guidance. In the broader context of global efforts to achieve the United Nations SDGs, this research paper contributes to the ongoing dialogue on sustainable development by providing quantitative results within the BRICS countries. By highlighting the impact of different drivers in shaping life expectancy, this study underscores the necessity of broader policy frameworks that prioritize economic progress and the well-being of citizens.

## 6. Policy recommendations

The results demonstrate that life expectancy in the BRICS is declining due to $CO_2$ emissions. The BRICS countries need to take necessary steps in the future to implement policies that reduce pollution and enhance health outcomes. Firstly, they should start implementing and enforcing strict emission control measures across all industry sectors, including setting emissions limits and reduction targets. Such regulatory frameworks should be supplemented by robust monitoring mechanisms and punitive measures for non-compliance, engendering a salient incentive for emission reduction. Secondly, it is necessary to foster a transition towards greener energy involving the augmentation of capacities of alternative energy and the phased reduction of fossil fuel reliance. Promotion of R&D efforts and offering incentives can accelerate this shift. The third way to minimize $CO_2$ emissions is through awareness efforts highlighting the health risks of prolonged $CO_2$ exposure. Furthermore, these policies can have a multiplicative effect on one another. With the help of interdisciplinary partnerships and global cooperation, they can diminish the detrimental effect of $CO_2$ emissions on longevity, thereby manifesting a salutary effect on the public's health.

On the other hand, GDP had a positive impact on life expectancy and the coefficient is 0.639***. More progress in economic growth and GDP-generating sectors is needed to further increase life expectancy in the future [109]. Firstly, strengthening GDP can strengthen medical and healthcare systems, upgrade medical facilities, and broaden the accessibility of modern healthcare services. Additionally, prioritizing equitable wealth distribution guarantees that the advantages of economic prosperity are accessible to all segments of society, fostering enhancements in living standards and facilitating healthcare access. Furthermore, fostering comprehensive educational initiatives on health, hygiene, and preventive care, supported by an upward trend in GDP, empowers individuals to make well-informed decisions regarding their lifestyle choices conducive to better health. Also, it is imperative to allocate resources toward studies and development endeavours to tackle prevailing health issues and advance groundbreaking medical interventions. Collaborative efforts between governmental bodies, private enterprises, and civil society can synergistically leverage boosted GDP to propel the advancement of public health initiatives. A judicious confluence of these measures, underpinned by a commitment to equitable development, holds promise in capitalizing on the affirmative association between GDP and life expectancy, culminating in enhanced overall welfare and societal advancement.

To capitalize on the empirically documented positive association between renewable energy utilization and life expectancy in BRICS, crafting practical policy recommendations is significant in harnessing this relationship while boosting public health. Firstly, fostering environment conducive to mainstream clean power necessitates implementing targeted incentives and regulatory structures. These mechanisms can encompass tax incentives, subsidies, and

streamlined permitting processes to facilitate the integration of energy-efficient technologies. Furthermore, it is crucial to prioritize initiatives to conduct research and development in green power production. Financial resources can facilitate progress in power storage, grid integration, and efficiency improvements, promoting the development of a stronger and more resilient renewable energy landscape. Furthermore, it is imperative to emphasize the significance of public understanding and educational efforts regarding the health advantages associated with renewable energy sources. Citizens with knowledge and awareness can actively support and promote laws about renewable energy, diminishing their dependence on fossil fuels and ameliorating the adverse health consequences associated with such reliance. Collaborative partnerships between governmental entities, private sectors, and research institutions can catalyze the adoption of renewable energy sources, contributing to prolonged life expectancies and improved public health.

Given the mixed relationship between trade and life expectancy, it is advisable to propose policy recommendations to mitigate the potentially unfavourable consequences. To begin with, it is imperative to develop a sophisticated approach toward trade policies that highlights the integration of health considerations in conjunction with economic aims. Striking a balance between economic prosperity and public health necessitates the formulation of trade agreements that incorporate stringent ecological and health standards, thus alleviating the potential negative externalities on life expectancy. Secondly, investing in robust healthcare infrastructure and universal access to quality healthcare services assumes paramount significance. The revenue generated from trade activities could be channelled towards strengthening healthcare systems, remodelling medical facilities, and augmenting health services accessibility, thereby increasing the overall well-being of citizens. Cross-sectoral collaboration between trade and health ministries is instrumental in aligning policy priorities and ensuring that trade openness aligns with health requirements. Eventually, a harmonized and comprehensive approach to trade policies, underpinned by a firm commitment to public health enhancement, holds promise in counteracting the observed detrimental influence of trade openness on life expectancy, thereby fostering more equitable and sustainable socioeconomic development.

## 7. Limitations and future research

Despite the valuable contributions of this research, several limitations need to discuss. Firstly, the study's focus on the BRICS countries may limit the generalizability of findings to other economies. Secondly, the analysis primarily utilizes aggregate data, potentially overlooking subnational variations that could affect the observed relationships. Additionally, the PMG-ARDL approach assumes homogeneity of coefficients across countries, potentially neglecting unique country-specific dynamics.

Future research will extend and refine this study in various ways. Firstly, exploring individual country analyses within the BRICS could unveil nuanced variations in the identified relationships. Secondly, incorporating diverse data, such as region-specific indicators or sectoral contributions, may improve the accuracy of the analysis. Thirdly, a longitudinal approach could offer insights into the dynamic evolution of the examined variables over time. Moreover, expanding the study to encompass a broader spectrum of emerging economies would provide a more comprehensive understanding of the interplay between economic development, energy utilization, trade, and health outcomes.

## Supporting information

**S1 Appendix. List of abbreviations.**
(DOCX)

## Author Contributions

**Conceptualization:** Funda Kaya, Liton Chandra Voumik, Salma Akter, Sayeem Hasan Khan, Konrad Kochański, Grzegorz Zimon.

**Data curation:** Funda Kaya, Liton Chandra Voumik, Mamunur Rashid, Salma Akter, Sayeem Hasan Khan.

**Formal analysis:** Funda Kaya, Liton Chandra Voumik.

**Methodology:** Salma Akter.

**Resources:** Funda Kaya, Liton Chandra Voumik.

**Software:** Salma Akter.

**Supervision:** Funda Kaya, Mahdi Salehi, Grzegorz Zimon.

**Writing – original draft:** Funda Kaya, Mamunur Rashid, Mahdi Salehi, Konrad Kochański, Grzegorz Zimon.

**Writing – review & editing:** Funda Kaya, Mamunur Rashid, Mahdi Salehi, Konrad Kochański, Grzegorz Zimon.

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
