## [Decision Letter · Decision Letter 0]

28 Jun 2024

PONE-D-24-16740Energy Choices to Health Outcomes: A Multidimensional Analysis of Risk in BRICS via PMG-ARDL ApproachPLOS ONE

Dear Dr. Zimon,

Thank you for submitting your manuscript to PLOS ONE. After careful consideration, we feel that it has merit but does not fully meet PLOS ONE’s publication criteria as it currently stands. Therefore, we invite you to submit a revised version of the manuscript that addresses the points raised during the review process.

We look forward to receiving your revised manuscript.

Kind regards,

Abid Rashid Gill

Academic Editor

PLOS ONE

3. We note that you have referenced (ie. Barro, R. J.  et al. [73]) which has currently not yet been accepted for publication. Please remove this from your References and amend this to state in the body of your manuscript: (ie “Barro, R. J. et al. [Unpublished]”) as detailed online in our guide for authors

Additional Editor Comments:

The paper need to highlight the research gap and significance of the study. Moreover, theoretical framework should justify the direct and indirect relation between health outcome and energy consumption and carbon emission.

Reviewers' comments:

Reviewer's Responses to Questions

**Comments to the Author**

1. Is the manuscript technically sound, and do the data support the conclusions?

Reviewer #1: Yes

Reviewer #2: Yes

2. Has the statistical analysis been performed appropriately and rigorously? 

Reviewer #1: Yes

Reviewer #2: Yes

3. Have the authors made all data underlying the findings in their manuscript fully available?

Reviewer #1: Yes

Reviewer #2: Yes

4. Is the manuscript presented in an intelligible fashion and written in standard English?

Reviewer #1: Yes

Reviewer #2: Yes

5. Review Comments to the Author

Reviewer #1: The current study based on energy choices to health outcomes is really a need of the day. Problem statement is well aligned with research questions, objectives, literature review, methodology and findings of the current study. Life expectancy as proxy of health outcomes is well established variable. Date collected from BRICS countries can be beneficial for rest of the countries as well. The policy makers from all over the world can use these findings to make polices to mitigate the degrading effect of climate change, especially in emerging economies.

Reviewer #2: The following revisions are suggested.

1. It is suggested to revise the title. Otherwise, justify the use of these terms: Energy Choice, Multidimensional Analysis, Risk, with the contents of the study.

2. Add one or two sentences, in the very beginning of the abstract, narrating the significance of the study.

3. Several studies have already evidenced health outcome as a function of economic growth, energy consumption, and carbon-emission. At the same time, many studies have revealed the economic-environmental-social interdependence in shaping the development trajectory. What is the novelty of the current study?

4. Economic growth-health outcome is quite logical and empirically testified as well. Energy consumption and carbon dioxide emission are directly linked to economic growth (through the energy-growth-emission model), hence, indirectly with the health outcome. It seems that the current study has fitted the energy-growth-emission model for the health outcome. It is logical that energy consumption (REC and NREC) in developing economies leads to economic growth, and due to more fossil fuel use (for economic growth) carbon dioxide emission is obvious. So, for the current study, could there be a way to separate the direct and indirect effects of Energy consumption and carbon dioxide emission on the health?

5. What is the purpose of presenting the Dumitrescu Hurlin panel causality, especially, where the causality moves from life expectancy towards carbon dioxide emissions and towards fossil fuel consumption.

6. The study has used the PMG-ARDL and the FMOLS tests. The FMOLS test was used to test the robustness of the PMG-ARDL test results. What is the implication of varying signs of the coefficient logTO in both tests?

7. The place for the Figure 1 is mentioned but the figure is nowhere in the manuscript.

8. A citation “Rahman, M. M., & Alam, K. (2022)” is numbered differently, i.e., as 42 and then as 101 (verify in the complete references).

6. PLOS authors have the option to publish the peer review history of their article (what does this mean?). If published, this will include your full peer review and any attached files.

Reviewer #1: **Yes: **Mubasher Ishfaq

Reviewer #2: **Yes: **Tusawar Iftikhar Ahmad

---

## [Author Response · Author response to Decision Letter 0]

6 Aug 2024

Dear Reviewers, 

We made corrections according to the reviewers' suggestions. Details are color-coded in the article. Each reviewer has a different color

We hope that our article is now better and has a chance of being published.

Sincerely

Grzegorz Zimon

The following revisions are suggested.

1. It is suggested to revise the title. Otherwise, justify the use of these terms: Energy Choice, Multidimensional Analysis, Risk, with the contents of the study.

Response: Thanks Professor, we revised the title. 

2. Add one or two sentences, at the very beginning of the abstract, narrating the significance of the study.

Response: Thanks, professor. We have provided 2 lines at the beginning of the abstract, that narrate the significance of the study.

3. Several studies have already evidenced health outcomes as a function of economic growth, energy consumption, and carbon emissions. At the same time, many studies have revealed that economic-environmental-social interdependence in shaping the development trajectory. What is the

the novelty of the current study?

Response: Thanks, Professor. We have provided the novelty of this paper and marked its by purple color.

4. Economic growth-health outcome is quite logical and empirically testified as well. Energy consumption and carbon dioxide emissions are directly linked to economic growth (through the energy-growth-emission model), hence, indirectly with the health outcome. It seems that the current study has fitted the energy-growth-emission model for the health outcome. It is logical that energy consumption (REC and NREC) in developing economies leads to economic growth, and due to more fossil fuel use (for economic growth) carbon dioxide emission is obvious. So, could there be a way to separate the direct and indirect effects of Energy consumption and carbon dioxide emission on the health?

Response: Information about the relationship between energy consumption and CO2 emissions was written in the content of the manuscript.

5. What is the purpose of presenting the Dumitrescu Hurlin panel causality, especially, where the causality moves from life expectancy towards carbon dioxide emissions and fossil fuel consumption.

Response: Thanks, Professor. We have provided the purpose of presenting the Dumitrescu Hurlin panel causality, especially, where the causality moves from life expectancy towards carbon dioxide emissions and fossil fuel consumption and is marked by purple color.

6. The study has used the PMG-ARDL and the FMOLS tests. The FMOLS test was used to test the robustness of the PMG-ARDL test results. What is the implication of varying signs of the coefficient logTO in both tests?

Response: Thanks, Professor. We have provided the implication of varying sign of the coefficient logTO in PMG-ARDL and FMOLS.

7. The place for the Figure 1 is mentioned but the figure is nowhere in

the manuscript.

Response: Thanks, Professor. We have added the figure.

8. A citation "Rahman, M. M., & Alam, K. (2022)" is numbered differently, i.e., as 42 and then as 101 (verify in the complete references).

Response: Thanks, Professor. We have edited the references.

---

## [Decision Letter · Decision Letter 1]

3 Sep 2024

Energy Choices to Health Outcomes: A Multidimensional Analysis of Risk in BRICS via PMG-ARDL Approach

PONE-D-24-16740R1

Dear  Authors,

We’re pleased to inform you that your manuscript has been judged scientifically suitable for publication and will be formally accepted for publication once it meets all outstanding technical requirements.

Kind regards,

Abid Rashid Gill

Academic Editor

PLOS ONE

Additional Editor Comments (optional):

Reviewers' comments:

Reviewer's Responses to Questions

**Comments to the Author**

1. If the authors have adequately addressed your comments raised in a previous round of review and you feel that this manuscript is now acceptable for publication, you may indicate that here to bypass the “Comments to the Author” section, enter your conflict of interest statement in the “Confidential to Editor” section, and submit your "Accept" recommendation.

Reviewer #2: All comments have been addressed

2. Is the manuscript technically sound, and do the data support the conclusions?

Reviewer #2: Yes

3. Has the statistical analysis been performed appropriately and rigorously? 

Reviewer #2: Yes

4. Have the authors made all data underlying the findings in their manuscript fully available?

Reviewer #2: Yes

5. Is the manuscript presented in an intelligible fashion and written in standard English?

Reviewer #2: Yes

6. Review Comments to the Author

Reviewer #2: (No Response)

7. PLOS authors have the option to publish the peer review history of their article (what does this mean?). If published, this will include your full peer review and any attached files.

Reviewer #2: **Yes: **Tusawar Iftikhar Ahmad

---

## [Editor Report · Acceptance letter]

15 Sep 2024

PONE-D-24-16740R1 

PLOS ONE

Dear Dr. Zimon, 

I'm pleased to inform you that your manuscript has been deemed suitable for publication in PLOS ONE. Congratulations! Your manuscript is now being handed over to our production team.

Kind regards, 

on behalf of

Dr. Abid Rashid Gill 

Academic Editor

PLOS ONE